# A Review on the Usage of Continuous Carbon Fibers for Piezoresistive Self Strain Sensing Fiber Reinforced Plastics

Patrick Scholle * and Michael Sinapius

Institute of Mechanics and Adaptronics, Technische Universtität Braunschweig, Langer Kamp 6, 38106 Braunschweig, Germany; m.sinapius@tu-braunschweig.de
* Correspondence: p.scholle@tu-braunschweig.de

**Abstract:** This literature review examines the application of carbon fibers and their reinforced plastics for Self-Strain-Sensing structures and gives an up-to-date overview of the existing research. First, relevant basic experimental approaches that can be found in the literature are presented and discussed. Next, we propose to cluster the available articles into 5 categories based on specimen size and ranging from experiments on bare carbon fiber via impregnated fiber rovings to carbon fiber laminates. Each category is analyzed individually and the potential differences between them are discussed based on experimental evidence found in the past. The overview shows, that the choice of carbon fiber and the specific experimental setup both significantly influence the piezoresistive properties measured in Self-Strain-Sensing carbon fiber reinforced plastics. Conclusively, based on the conclusions drawn from the literature review, we propose a small number of measurements that have proven to be important for the analysis of Self-Strain-Sensing carbon fiber structures.

**Keywords:** self sensing; piezoresistivity; carbon fiber; electrical resistance



## 1. Introduction

Self Sensing is generally referred to as a way to monitor some property of a physical object without the necessity to attach an additional discrete sensor to it. Examples for this can be found in piezoelectric based devices that combine both sensor and actuator into a single piezoelectric ceramic, thus combining two functions in one entity. Another popular field for Self Sensing is the structural monitoring of fiber reinforced plastics. In this application, the two functions of load-carrying and strain or damage sensing are fulfilled by carbon fibers simultaneously.

Conventionally, sensors are attached to the surface of a part to measure strain. A variety of measurement principles have been developed over the years that fulfill this task. Strain gauges are the most commonly used tools for this purpose and have been extensively studied since their invention in the 1940s. Attached sensors can influence the shape of a surface and encounter problems in abrasive environments. Additionally, only informations about the part surface can be acquired. More recently in the field of composite materials, researchers started to integrate sensors into the part to overcome this problem. Fiber-based sensors are a popular choice in this field due to their small size and precise measurement. The manufacturing and usage of functional fibers that can fulfill a large number of tasks such as strain sensing, energy storage, energy harvesting and more has been studied extensively and more and more functional principles have been discovered in the past [1]. However, when sensors are integrated into a structure, a material inconformity arises due to dissimilar mechanical properties of the materials such as youngs modulus and thermal expansion coefficient. This inconformity can result in strain concentrations at the interface of material and sensor and lead on to a detachment of the sensor and thus to a loss or deterioration of sensing ability. Furthermore, the detached Sensor might lead to delaminations that ultimately cause the failure of the structural part. These thoughts

led to a continuous effort to miniaturize integrated sensors, thus minimising the disturbance caused by the foreign material. This approach however generates a more and more localized measurement and thus limits the possibility to measure global responses.

Among other reasons, this predicament led researchers to the idea of Self-Sensing structures towards the end of the 20th century. Self-Sensing was inspired by researchers asking questions about the necessity to use a foreign material to monitor strain and damage of a structure. What if the structural material itself could provide information about its current strain and damage? Strain and damage monitoring would no longer be achieved in an indirect manner. Instead of analyzing strain and strain rearrangements within the part by different sensors at discrete locations, a property of the structural part itself could be measured directly. Carbon Fiber Reinforced Plastics (CFRPs) were found to offer this ability and numerous research was inspired by this finding. Researchers found ways to correlate strain, damage and temperature within a laminate to the electrical impedances of the load carrying structural fibers themselves. Mechanical strain alters the electrical resistance due to dimensional and piezoresistive effects reversibly [2]. Temperature changes the resistivity of the semiconducting material with a negative temperature coefficient [3]. Fiber rupture alters the resistance irreversibly by deleting conduction paths [4]. Delaminations and matrix cracks within the material alter the resistive, capacitive and inductive transmission of electrical current [5–7]. Electrical measurements on the load carrying carbon fibers themselves can therefore be used to generate a global, integrated health and usage monitoring system with ideal structural conformity [8]. This direct measuring of the load carrying structure and the ideal structural conformity could prove to be advantageous in the future when compared to approaches working with discrete sensors made from a foreign material. Possible applications of this idea are manifold and different applications have been proposed in the past. For example, Self-Sensing carbon fibers have been proposed to be used as a health monitoring technology for large bridges [9], X-Ray transparent deflection monitoring of composite beams [10] and load and failure detection in composite aircraft wings [8]. Furthermore, resistance measurements have also been proven to be helpful in material testing, where resistance measurements have been used to characterize the failure of single carbon fibers under compression loads [11].

In this review, we discuss experimental works that attempt to measure mechanical strain using Self-Sensing approaches. It has been shown in the past that ambient changes in humidity and especially temperature have a significant impact on the measured resistance of carbon fiber based sensors [3,12]. While these things have to be considered for practical implementations outside the laboratory environment, we exclude these effects in the following discussion in order to limit the complexity of the analysis. Other great review articles have been written in the past that discuss a subset of articles analyzed in this work. However, these review articles often work on a broader scope, e.g., reviewing multifunctional polymer-matrix composites in general [13,14], or both strain and damage monitoring of CFRP laminates [15]. This article is dedicated to the specific field of strain sensing in order to more thoroughly discuss research for this application and is aimed to give an up-to-date overview of this field. There is a large number of articles working in this field that have not been communally compared yet, possibly because they work on different types of specimen. For example, some research articles study the piezoresistive properties of single, bare, carbon fibers, while other articles discuss piezoresistive properties of large carbon fiber laminates. We believe that we can learn a lot from analyzing and comparing results between all of these different types of experiments, finding general principles that occur everywhere and trying to explain the reasons for differences in results. To achieve this task, we novelly propose to cluster the existing research into 5 categories of growing part complexity which we believe to be helpful in understanding the functioning principle of a carbon fiber based sensor:

1.  The simplest level of complexity we analyze in this work is that of a single carbon fiber filament under uniaxial strain. Numerous experiments were conducted that analyze the change of electrical resistance of single carbon fibers due to mechanical strain.
2.  The next step in complexity arises by adding a polymer matrix to the fiber.
3.  The complexity of the system is larger when multiple filaments are embedded into a polymer matrix in the form of a roving.
4.  Multiple rovings form a ply of carbon fiber. The current is then distributed between significantly more filaments
5.  The last step of complexity regarded in this article is that of a larger carbon fiber laminate consisting of multiple plies. Different layer orientation and resin rich interfaces can change the behavior of these parts when compared to single plies.

In addition to the different levels of complexity, experiments differ in electrical connection type, measuring principle, and other factors. Some of these factors are displayed in Figure 1 and will be discussed throughout this review.

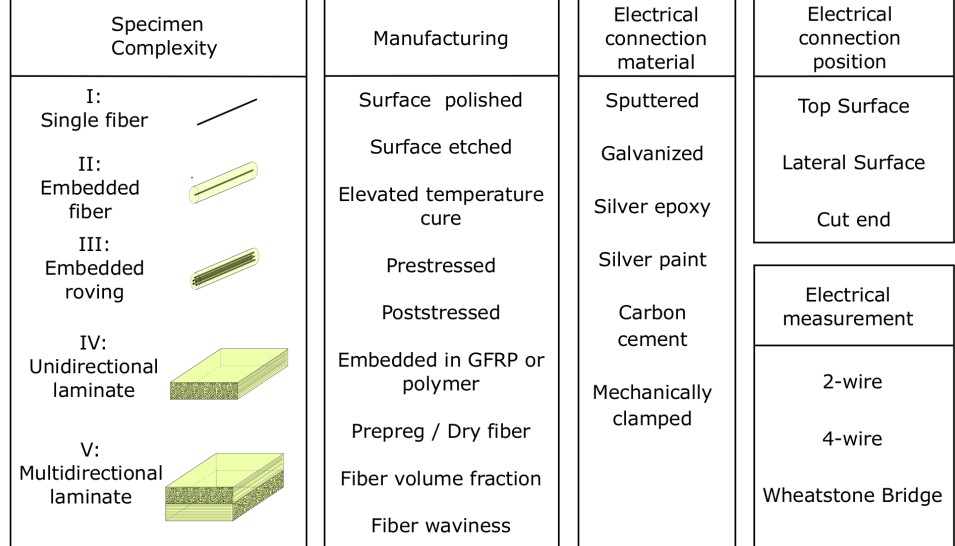

**Figure 1.** Overview of different experimental possibilities used in the past for the evaluation of Self-Strain-Sensing carbon fiber reinforced plastics.

We systematically searched for available literature within the scope of this review using a two-step approach. First, we searched for available literature dealing with strain monitoring using continuous carbon fibers based on typically used keywords. Figure 2 shows an overview of more than 100 articles published within the broader field of Self-Sensing carbon fiber reinforced plastics. The available articles found this way were then screened based on their title and abstract. Those articles that fall within the scope of this article were then selected for further review. In a second step, these articles were then used to find more articles by checking both the references within the article as well as other articles citing the work. This second step was then repeated for each new article. All articles found by this method are reviewed in this paper.

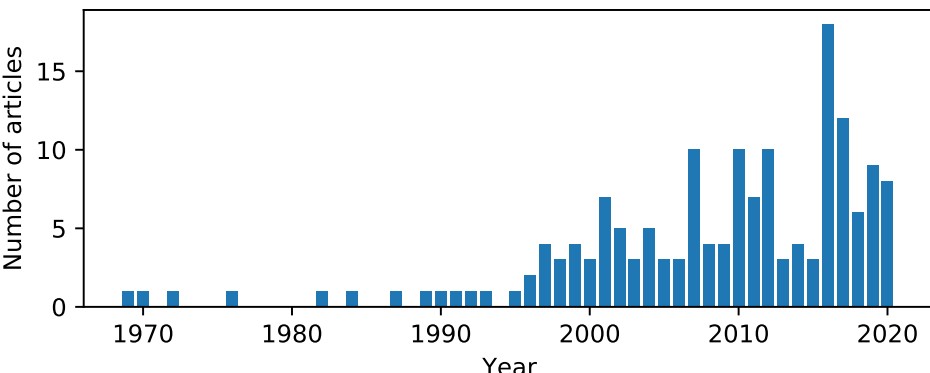

**Figure 2.** Overview of the published papers concerned with Self-Sensing properties of carbon fibers and their reinforced plastics.

## 2. Definitions and Mathematics of Strain Sensing

Before starting with the literature review, we would like to discuss some frequently used assumptions made in the literature. Articles working on strain induced resistance change often times fundamentally build upon the same principles. Authors assume, that the electrical volume resistance of a material $R$ can be calculated with [16] (p. 33):

$$R = \rho \frac{L}{A} \tag{1}$$

In this equation, $\rho$ is the electrical volume resistivity of the material, $L$ is the length of the conductor and $A$ is its cross section. Figure 3 (left) shows an example for an experimental setup where $A \hat{=} bh$. Some or all of these parameters can vary when the material is strained. Building the total differential of the equation, we can analyze the influence of each parameter to the change in resistance. When a round cross section $A = \pi r^2$ and a uniaxial load case is assumed, we can write:

$$\frac{dR}{R} = \frac{dL}{L} - \frac{dA}{A} + \frac{d\rho}{\rho} = \frac{dL}{L} - 2\frac{dr}{r} + \frac{d\rho}{\rho} \tag{2}$$

For uniaxial load, free boundary conditions and isotropic materials, the radial strain $\frac{dr}{r} = \epsilon_r$ is connected to the longitudinal strain $\frac{dL}{L} = \epsilon_L$ through the Poisson ratio $\nu$. The equation can then be rewritten:

$$\frac{dR}{R} = \epsilon_L(1 + 2\nu) + \frac{d\rho}{\rho} \tag{3}$$

The resistance change is thus dependent on two summands, one describing geometrical changes and one describing changes in resistivity. In this article, we will distinguish between these two factors by referring only to the change in resistivity as *piezoresistivity*. Other effects are described to be caused by *geometrical change*. For metallic materials, piezoresistivity is often times negligibly small. Thus, the resistance change due to strain is often times regarded to be strictly proportional with the proportionality constant referred to as gauge factor $GF = 1 + 2\nu$. Many semiconducting materials like silicon or germanium can undergo large changes in resistivity, that can be 50–100 times larger than the geometric term [17] (p. 2112). It will be discussed in this article if this is also the fact for carbon fibers. The major Poissons ratio of many carbon fibers is 0.2 [18]. If piezoresistivity is not taken into account, the gauge factor for these carbon fibers should therefore be equal to $GF = 1 + 2\nu = 1.4$. Deviations from this value can either be explained by errors in the measurement of the Poissons ratio, piezoresistive effects or the non-applicability of Equation (1).

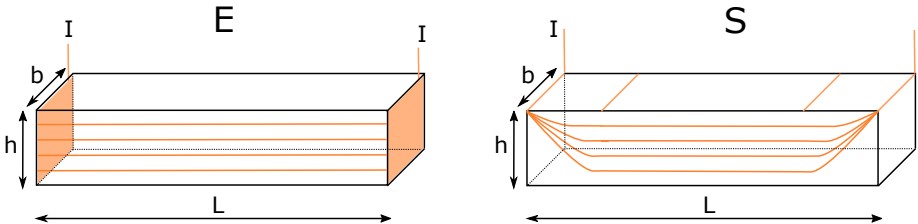

**Figure 3.** Current inhomogeneity in the cross sction as a possible explanation for differing results between measurements with end-contact and surface-contact.

### 3. Practical Implementations of Resistance Measurements

The majority of articles analyzed in the following review analyze the same physical quantity: The electrical resistance at DC currents. In practical implementation, this measurement is however done in a number of different ways. As will be shown, this is especially true for experimental setups analyzing rather large carbon fiber laminates. Furthermore, a large number of different manufacturing strategies for CFRP exists that can also have an influence on the behavior as a sensor. As will be further detailed in this review, the measurement setup and material plays an important role in determining the electrical resistance. In this section, we therefore present the possibilities that can be found in the literature for measuring the electrical resistance of carbon fibers and their reinforced plastics.

#### 3.1. Electrical Test Setups

A number of different test setups for measuring electrical resistance are reported. Many researchers use bridge circuits to measure the rather small changes in resistance occuring under mechanical load. The most commonly used bridge circuit is the Wheatstone Bridge, that has for many years been applied extensively in strain gauge experiments. If no bridge circuit is used, the resistance is often times measured based on Ohms law by using a constant current source and measuring the voltage drop across the specimen with a high-precision analog-digital-converter. For these measurements, 2-wire and 4-wire setups have to be differentiated.

In 2-wire measurements, the specimen is connected by two test leads. Cable and contact resistances are therefore in series to the resistance that is measured. This can lead to erroneous measurements, especially when the resistance to be measured is small. When small resistances are to be measured, contact resistances and cable resistances can significantly influence the measurement result. This is especially problematic if one of these resistances changes during the measurement, since it is impossible to distinguish between the resistance change of the material studied and the resistance change of the contact. One example for this would be a changing contact resistance due to straining the specimen. For this reason, 4-wire measurements are preferred in many cases, especially when small resistances are to be measured [19]. The basic idea behind this approach relies on two extra test wires that are used to measure the voltage drop across the resistance. Thus, two wires are used for current introduction and two isolated wires are connected to a high impedance voltmeter. With this setup, cable and contact resistances can be excluded from the measurement because only an insignificant current flows through the voltage leads.

#### 3.2. Contacting Carbon Fiber Laminates

After a test setup is chosen, every researcher attempting to measure the electrical resistance of CFRPs is confronted with a simple, yet in some ways complex question: How do I connect my measurement equipment to the carbon fibers? Throughout the years, researchers tried different ways to fulfill this task. A good overview is presented in [20] (p. 257ff). The most simple setups rely on mechanical clamping with metallic clamps. In these cases, the surface roughness of the carbon fiber part can result in inhomogeneous

contact and thus large contact resistances as well as inhomogeneous current introduction. To reduce this influencing factor, many experiments use some sort of conductive adhesive to connect the metallic wires of the measuring instrument to the fibers. In these adhesives, conductive particles are bonded by a binder material. Different combinations of particle and binder exist. Examples for adhesives are:

- Silver filled epoxies from various manufacturers. These epoxies form a mechanically durable bond with a relatively large conductivity(e.g., used in [21])
- Silver paint. Instead of epoxy, an organic binder with large concentration of solvent is used here. While the conductivity is similar to silver epoxies, the connection is not as mechanically durable (e.g., used in [22])
- Graphite cement. Graphite dust can be used instead of silver to generate conductivity. A popular binder material for low-cost applications is polyvenylacetat (PVA). The conductivity is generally lower than products based on silver (e.g., used in [21])

Another approach is the deposition of metal to the substrate surface. For example, metals can be deposited on the fibers surface through a galvanic deposition process. Nickel and Copper are among the most commonly used metals for deposition onto carbon fibers. It is either possible to galvanize the fibers before impregnation directly, or to galvanize the surface of the finished part. The latter possibility requires an appropriate surface modification for removing any resin rich surface zone and expose the conductive fiber. Another possibility for metal deposition onto parts surface is the sputtering technique. Once a metal is deposited to the surface, a typical electrical connection process can be used: Soldering. With metal deposition techniques, general purpose solder alloys can wet the fiber surface and allow direct soldering [23]. Carbon fibers typically cannot be directly soldered due to a lack of wetting. Burda et al. [24] showed a possible way to allow for a direct soldering of carbon based materials through transition metal rich alloys. To our knowledge, this approach is however not yet widespread used and tested thoroughly.

After answering the question of how to connect the measuring instrument to the fibers surface, the next question is where to attach this connection. In the case of rather large carbon fiber parts, two different geometrical contact variants displayed in Figure 3 have to be differentiated. End contacts are connected to the cut end of the specimen. Surface contacts on the other hand are attached to the lateral surface or part of the lateral surface. Both of these variants can be combined in 4-wire measurements. At first glance, the difference between these two setups can appear to be small. However, the current flow within the specimen can in fact be quite different in both cases. In the case of surface contacts, the current flow is two- or three-dimensional near the point of current introduction. Figure 3 (right) depicts schematically how current is rearranged through a 2-dimensional current flow in the cross section with growing distance from current entry. As Zimney et al. [25] point out in their article, this behavior is especially relevant for electrically anisotropic conductors such as carbon fiber reinforced plastics. Current non-uniformities have a large effect on the measured potential in the vicinity of the current electrode and could therefor also be relevant to the Self-Strain-Sensing properties of a specimen.

## 4. Resistance Change of Single Carbon Fibers due to Mechanical Strain

### 4.1. Examinations of Single Bare Carbon Fibers Under Tensile Load

The overall results of single fiber experiments are displayed in Table 1. Reported gauge factors from single fibers range from negative values of $-8.9$ to positive values of 5.1. Notably, an overall trend can be observed that small and negative gauge factors tend to be measured on carbon fibers with larger moduli. As dicussed with Equation (3), this means that a piezoresistive behavior—in this case negative piezoresistivity—is observed in these fibers. As displayed in the table, both 2-wire and 4-wire measurements have been used in the past. Since no obvious quantitative difference between these two experimental setups can be identified, we choose to review all of the available research on this subject. Experiments using 2-wire setups however have to be analyzed carefully as discussed in Section 3.1.

**Table 1.** Literature review of electromechanical coupling in dry carbon fiber.

| Manufacturer | Fiber Name | E / GPa | Gauge Factor | Method | Source |
|---|---|---|---|---|---|
| | undefined | 380 | 0.7 | 2-wire | [26] |
| | undefined | 340 | 1.3 | 2-wire | [26] |
| | undefined | 230 | 1.7 | 2-wire | [26] |
| Whitaker | Type 2 | 330 | 0.6 | 2-wire | [27] |
| Celanese | ? | 786 | not reproducible | 2-wire | [27] |
| Toray | T1000 | 300 | 2.0 to 2.2 | 2-wire | [11] |
| Grafil | 33-500 | 230 | 2.0 to 2.2 | 2-wire | [11] |
| Hercules | AS4 | 230 | 2.0 to 2.2 | 2-wire | [11] |
| Amoco | T650-35 | 240 | 2.0 to 2.2 | 2-wire | [11] |
| Hercules | AS4 | 231 | 2.02 | 2-wire | [28] |
| Amoco | P-25 | 170 | 1.14 | 4-wire | [29] |
| Amoco | P-75 | 520 | 0 | 4-wire | [29] |
| Amoco | P-100s | 650 | $-8.9$ | 4-wire | [29] |
| Amoco | P-120 | 830 | $-4.65$ | 4-wire | [29] |
| Homemade | PAN 1400 | 200 | 1.26 | 4-wire | [29] |
| Homemade | PAN 2000 | 240 | 0.82 | 4-wire | [29] |
| Homemade | PAN 2300 | 270 | 0.68 | 4-wire | [29] |
| Homemade | PAN 2900 | 340 | $-0.6$ | 4-wire | [29] |
| Celanese | GY-80 | 640 | $-3.87$ | 4-wire | [29] |
| Amoco | P-75/CuCl2 | 273 | $-1$ | 4-wire | [29] |
| Amoco | P-75/MnCl2 | 347 | $-3.1$ | 4-wire | [29] |
| Amoco | P100s/MnCl2 | 337 | $-7.3$ | 4-wire | [29] |
| Toray | T300 | 221 | 1.8–2.3 | 4-wire | [30] |
| Toray | T700S | 230 | 5.1 | 2-wire | [31] |
| Toray | T800H | 294 | $-2.2$ | 2-wire | [32] |
| Toho Tenax | HTA5241 | 238 | 1.86 | 2-wire | [33] |
| Hexcel | IM7 | 250 | $1.3 \pm 0.1$ | 4-wire | [34] |
| Toho Tenax | HM35 | 345 | 1.77 | 2-wire | [35] |
| Toray | T700 | 230 | $1.38 \pm 0.064$ | ? | [36] |

Conor and Owsten [26] are the first to study the piezoresistive effect of carbon fibers. The authors show that carbon fibers with highly oriented graphene layers show a smaller gauge factor than do lesser oriented fibers. They discuss the possibility of crystallite reorientation during straining as a possible explanation for this. Due to the high anisotropy of electrical conductance in a graphite crystal, reorientation could have a significant influence on the resistance. As has been shown by Curtis et al. [37], crystallites with the greatest misorientation undergo the greatest reorintation during straining. The authors argue that, when no volumetric change is assumed and no change of resistivity is apparent, the gauge factor of the material should be 2. In this reasoning, any gauge factor other than 2 is thus attributed to changes in the resistivity of the material. An increase in crystallite orientation would decrease the resistivity and, therefor, lower the observable gauge factor. Keeping the reorientation effects in mind, the gauge factor of low initial orientation fibers should therefor be smaller than for high initial orientation fibers. The authors are however not able to support this claim with evidence. In fact the authors observe the exact opposite. Thus, the authors argue, crystallite reorientation cannot be the main factor for the change in resistance.

Berg et al. [27] take these findings and expand the knowledge by analyzing two different fibers, namely a low modulus type and a high modulus type. In their experimental setup, a single carbon fiber is glued onto a card board frame (Figure 4). Electrical contact is achieved through silver paint. The cardboard frame is then mounted into a tensile testing machine. Before straining the fiber, the cardboard is cut by slowly burning through it. The authors observe a gauge factor of 0.6 for the low modulus type. The high modulus fiber decreases in resistance with applied strain. However, the authors don't observe reproducible results for different fibers of the same high modulus type.

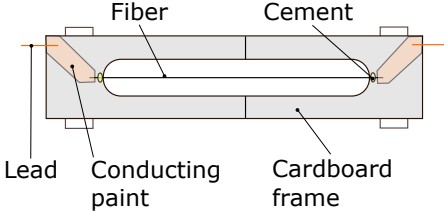

**Figure 4.** Experiment in [27].

Blazewicz et al. [29] analyze a large number of different carbon fibers both in-house manufactured and commercially bought with a 4-wire resistance measurement. They furthermore analyze the influence of intercalating fibers with metal chlorides ($MnCl_2$, $CuCl_2$) on the piezoresistivity. In accordance with previous authors, Blazewicz et al. [29] find that negative piezoresistivities play an important role in high modulus carbon fibers. The authors hypothesize, that an increase in carrier mobility due to a reduction in scattering mechanisms on the crystallite during stretching of the graphite fiber plays a dominant role in piezoresistivity. They support this theory by their experimental data measured on intercalated carbon fibers. Intercalated fibers show a significant decrease in resistivity due to an increase in charge carrier concentration.

Wang et al. [30] analyze the piezoresistive behavior of the T300 fiber using a 4-wire technique (Figure 5). They use silver paint for contacting the carbon fiber and report gauge factors between 1.8 and 2.3. The authors furthermore analyze quasistatic cyclic loading of the fiber and report on the reversibility of the resistance change. For low stress amplitudes of 18.8% of fracture stress, both strain and resistance change reversible upon unloading. For intermediate stress amplitudes of 58.1% the strain is still reversible, but the resistance increased irreversibly. For large stress amplitudes of 83% both strain and resistance increased irreversibly. This finding has not been reported before and could indicate damage mechanisms of the carbon fiber resulting in an irreversible increase in resistance.

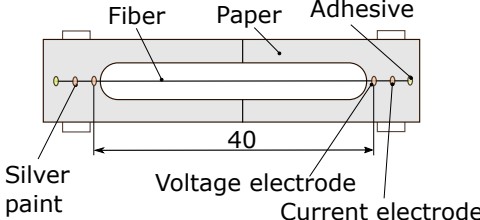

**Figure 5.** Experiment in [30].

**Discussion of single fiber experiments**

While there are various explanations for the exact reason of strain induced resistance change on single carbon fibers, the overall connections appear clear. Many carbon fibers have a positive gauge factor when being strained. This is especially true for PAN based carbon fibers with low to medium stiffnesses and high strengths (HS type). In many practical applications, it is these PAN based high strength carbon fibers that are being used due to their comparably small price, high strength and good availability. Depending on the exact fiber used, gauge factors generally range from 0.7 to 2.3. Keeping in mind the discussion of gauge factors with Equation (3), many of these values can largely be explained by the influence of dimensional change to the fiber. Other fibers within this class show different levels of positive or negative piezoresistive behavior. Piezoresistivity appears to vary between carbon fibers from different manufacturers, precursor type and manufacturing conditions. It has repeatedly been shown to be largest for ultra high modulus fibers, where it decreases the gauge factor. Overall,the results show that a linear relationship between strain and electrical resistance is measured for many carbon fibers.

### 4.2. Examinations on Single Carbon Fibers Embedded into Polymer

In practical applications, fibers are not used on their own. They are embedded into a polymeric matrix that, among other tasks, adhesively joins the fibers to one another, distributes the stresses among them, fixates the fibers in the required orientation and supports the fibers against buckling under compressive loads. The most popular polymer matrices are thermoset polymers. In many cases, elevated temperatures are used to increase the speed of the cross-linking reaction of the thermoset. It is widely accepted, that residual stresses form inside a composite material due to differences in thermal expansion of fiber and matrix, cure shrinkage and other factors [38] (p. 45ff). This thermal incompatibility might change the piezoresistive response of carbon fiber. Furthermore, the free boundary condition assumption used to develop Equation (3) is no longer valid when fibers are embedded in a matrix, because the integrated fiber is no longer allowed to freely change dimensions in accordance to its own Poisson ratio. This could also change the piezoresistive response of the carbon fiber. The results of this type of experiment are summarized in Table 2.

**Table 2.** Literature review of electromechanical coupling of embedded carbon fiber filaments.

| Manufacturer | Fiber Name | E/GPa | Matrix | Method | Initial Gauge Factor | Source |
|---|---|---|---|---|---|---|
| Hercules | AS4 | 231 | Epoxy Epon 828 | 2-wire | 2.02 | [28] |
| Toray | T300 | 230 | Epoxy Epon 9405 | 4-wire | $-17$ | [39] |
| Toray | T800H | 294 | PVA | 4-wire | $\approx -30$ | [40] |
| Toray | T700 | 230 | Epoxy | 4-wire | 1.61 | [41] |
| Toho Tenax | HM 35 | 345 | LM E20/H20 | 2-wire | 0.42 | [35] |
| Toray | T700 | 230 | Polyimid | 4-wire | $1.39 \pm 0.017$ | [36] |
| Toray | M55j | 540 | Polyimid | 4-wire | $0.88 \pm 0.035$ | [36] |
| Nippon | XN90 | 860 | Polyimid | 4-wire | $-3.97 \pm 0.175$ | [36] |

Crasto et al. [28] show initial experiments in this field. At the beginning of their research paper, the authors recreate single fiber experiments similar to those shown previously and measure a linear gauge factor of approximately 2 for AS4 fibers. Afterwards, they validate a different kind of experiment. The authors glue a single AS4 carbon fiber filament onto an acrylic beam and measure the resistive change under bending with a wheatstone bridge (WB) (Figure 6). This type of experiment allows to induce tension as well as compression to the fiber by bending the acrylic beam. They embed a single filament in an epoxy matrix and cured the system at room temperature. Straining the fiber after embedment again results in a linear gauge factor of approximately 2 for tension and small compressive strains. Both bare and embedded fiber thus show very similar gauge factors. Furthermore, the authors discuss the residual strain due to curing. As the authors explain, when a polymer matrix cures around a carbon fiber, resin shrinkage should induce radial and axial residual compressive stress. A cure at higher temperature should further increase the compressive stress due to differences in thermal expansion during cooldown. The authors observe an increase in resistance of approximately 0.5% with cure at room temperature. When cured at elevated temperatures of 83 °C, the specimen showed an even higher increase in resistance of 1.03%.

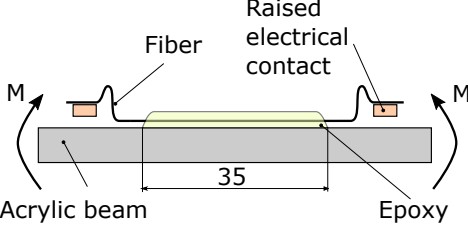

**Figure 6.** Experiment from [28].

Wang et al. [30,39] observe a similar increase of resistance during polymer cure in their work. The authors analyze a single Torayca T-300 fiber embedded in an epoxy resin and cured at 180 °C (Figure 7). The authors observe an even larger increase in resistance of 10%. They explain that this difference is due to the higher curing temperature resulting in a larger thermal shrinkage. On subsequent straining of the embedded fiber, the authors observe a decrease in resistance with a gauge factor of approximately −17 until strains of 0.5%. After this strain, the resistance starts to increase. The authors explain this phenomenon with an initial decrease of the residual compressive stress build up during curing at elevated temperatures. The authors attribute this finding to a newly discovered piezoresistive effect that they explicitly distinguish from the resistance change of non-embedded carbon fibers.

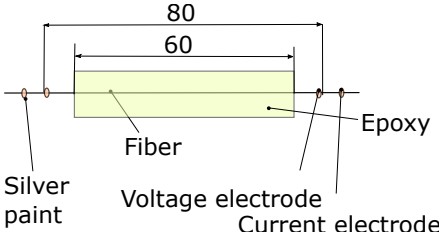

**Figure 7.** Experiment from [30].

Yoshitake et al. [40] analyze the electromechanical coupling of a single Toray T800HB fiber embedded into a soft polyvinyl acetate polymer under bending loads (Figure 8). The authors do not discuss the above mentioned resistance change during polymer curing. Instead, the authors report on a number of bending experiments similar to those of Crasto et al. [28] For compressive strains, the authors observe a decrease in electrical resistivity. For tensile strains, the authors also measure a decrease in resistivity, but only for stresses until approx. 0.3 GPa(equal to approx. 0.15%strain). The resistivity afterwards grows for larger stresses. The authors present an explanation that focuses on the electrical conduction mechanism involving the $\pi$ bonding electron in the graphite structure. Since a compressive loading decreases the layer distance of the graphite structure, the density of $\pi$ bonding electrons becomes larger. The authors argue, that this mechanism explains the reduction in resistivity in compression. In tension, the authors argue that dangling bonds of the graphite structure play a central role. They argue, that small strains increase the orientation and order in the graphite crystal. This is because the coulomb repulsive force of the dangling bonds distorts the unloaded graphite cell. Straining the structure thus increases order in the crystal because this source of distortion becomes less effective with more space between the dangling bonds. For higher stresses, the authors argue that the increasing distance between dangling bonds increases the resistance, thus explaining the apparent rise in resistivity. The authors furthermore find qualitatively similar results for non-embedded carbon fibers [32].

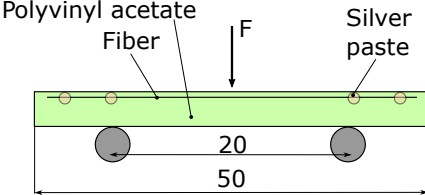

**Figure 8.** Experiment from [40].

Kalashnyk et al. [35] analyze both bare HM35 fiber as well as single carbon fibers embedded in an epoxy matrix. Furthermore, they analyze the same fiber in an unsized configuration which is achieved by heat treatment at 600 °C. Electrical measurements are performed with a 2-wire technique (Figure 9). In a novel approach, the authors use

Raman Spectroscopy to measure the mechanical strain of embedded carbon fibers. For non-embedded specimen, the authors do not observe significant differences for sized and unsized specimen in electromechanical response and calculate a mean gauge factor of 1.74 and 1.77 respectively. For embedded fibers, the authors are able to quantify the shrinkage induced compressive strain with the help of Raman measurements. According to their results, sized and unsized fibers have a residual fiber strain of −0.59% and −0.75% respectively. The authors furthermore measure the resistance change of single embedded carbon fibers under tensile load. Notably, the authors find that the gauge factor (when calculated with the fiber strain measured by Raman measurements) is smaller than that for non-embedded fibers. They measure values of 1.58 and 1.31 for sized and unsized specimen. For unsized specimen, the authors measure a strong decrease of resistance for small tensile loads. The authors hypothesize that this is not necessarily caused by the residual stresses, but could also be due to a lack of straightness of the unloaded specimen.

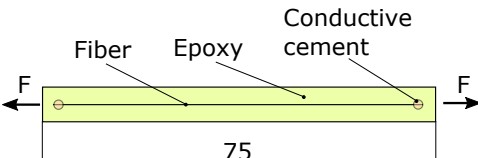

**Figure 9.** Experiment from [35].

Yao et al. [36] bond 3 different carbon fibers to an acrylic beam using polyimid. The polyimid layer is generated by spincoating a polyamic acid on the steel surface and subsequent heating procedure. The bending setup (Figure 10) is compared to a more classical approach using a cardboard frame and showed very good agreements. In accordance with previous authors, Yao et al. [36] state that low modulus carbon fibers show a negligible piezoresistivity, whereas higher modulus fibers show a more and more negative piezoresistive effect. The authors combine an electrical model based on Maxwell Garnett theory and a mechanical model developed by Northolt et al. [42]. In the electrical model, the carbon fiber is assumed to be composed of electrically anisotropic Basic Structural Units (BSUs) embedded in an isotropic host material. The authors hypothesize, that the volume fraction of BSUs and isotropic host play a critical role in the piezoresistive behavior of a carbon fiber. The authors calculate a volume fraction for each carbon fiber with an optimization routine and find good correlations between calculated and measured piezoresistivities.

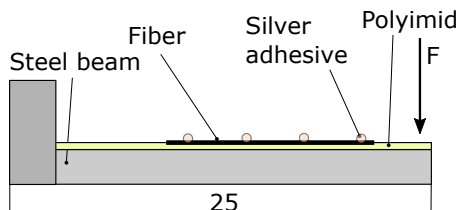

**Figure 10.** Experiment from [36].

**Discussion of embedded single fiber experiments**

In summary, some repeatable results have been reported for single carbon filaments cured in epoxy. There is an increase in resistance during polymer cure that is likely to originate from the radial and longitudinal compressive strain on the fiber [28,30].

Conflicting results are reported for the subsequent influence of longitudinal strain on the electrical resistance. In some cases, a decrease of resistance with small tensile strains is observed [35,39,40]. Yoshitake et al. [40] attribute this to a microstructural effect involving the graphite structure of the fibers that is independent on whether or not the fiber is embedded into a polymer. Wang et al. [39] explain that this behavior is due to thermal

shrinkage of the matrix and the resulting residual strain of the fiber. Kalashnyk et al. [35] attribute the initial decrease to a lack of straightness in the unloaded specimen. In other experimental reports, this decrease in resistance for small tensile strains is not observed at all [28,36]. Instead, a linear increase with gauge factors that are comparable to those of non-embedded fiber experiments are measured. This could however be caused by different manufacturing conditions or different materials. The piezoresistive behavior for carbon fibers under small tensile strains is therefor not clearly answered in the literature. More research analyzing this subject is necessary to clearly explain which of these explanations is true under which circumstances.

Larger tensile strains on the other hand are repeatedly reported to increase the electrical resistance. Compressive strains are similarly repeatedly shown to decrease a fibers resistance.

## 5. Embedding Carbon Fiber Rovings into a Polymer for Strain Sensing

While experiments on single embedded fibers are very interesting from a theoretical perspective, many practical setups for Self-Strain-Sensing structures use a multitude of carbon fibers simultaneously. These so-called rovings can be purchased with different number of filaments, typically ranging from 1000 to 24,000. This step from single to multiple fibers results in some significant changes in the experiments. Table 3 summarises the reported results obtained using carbon fiber rovings.

Residual stresses on the fibers due to thermal expansion incompatibility and polymer shrinkage are now distributed among multiple fibers. A simplified modelling approach used by Zobeiry et al. [38] (p. 47) shows that the resulting residual stresses are dependent on the fiber volume fraction. Thus, it seems reasonable to conclude that the influence of residual stresses on the individual fiber should be of lesser importance than in the case of a single carbon fiber.

The electrical current flow in a roving is more complex due to the large number of filaments. Under ideal assumptions, all fibers can be regarded to be connected in parallel to one another. In reality, transverse conduction processes originate from fibers in close contact and are thus a randomly distributed phenomenon in the specimen [43,44]. If the current is not introduced homogeneously to all fibers within the roving or a potential difference exists between two touching fibers for other reasons such as broken fibers, current will flow in between these fibers. As this conductive network can change due to straining of a roving, this could have a significant influence on the acquired resistance during straining.

**Table 3.** Literature review of electromechanical coupling of embedded carbon fiber rovings.

| Manufacturer | Fiber Name | Fila-Ment Count | Connection | Method | Gauge Factor | Source |
|---|---|---|---|---|---|---|
| Toray | T300 | 1 k | galvanic deposition | WB | 1.71 | [45] |
| Toho Tenax | HTA40 | 3 K | wound wire | WB | 1.3 | [46] |
| Toray | T700SC | 12 k | silver paint | WB | 2 | [47] |
| Toho Tenax | HTA40 | 1 k | clamped | WB | 1.72 | [48] |
| Toho Tenax | T300B | 1 k | clamped | WB | 1.54 | [48] |
| Nippon | PAN based | 6 k | silver paint | WB | 2.85–3.36 | [22] |
| Toray | T700SC | ? | ? | WB | 4–5 | [49,50] |
| unspecified | PAN based | 24 k | silver adhesive | WB | 1.35 | [51] |
| Toho Tenax | HM35 | 35 | carbon cement | 2-wire | 1.96–2.17 | [52] |
| Toho Tenax | T300B | 3 k | silver adhesive | 2-wire | 1.5 | [53] |

Horoschenkoff et al. [45] analyze unidirectional T300B 1k carbon fiber embedded in a glass fiber laminate (Figure 11 (top)). They apply a galvanic deposition process with subsequent soldering as contacts and a wheatstone bridge for measuring the resistance change. The authors measure a gauge factor of 1.71. They furthermore analyze the ex-pitch fiber Nippon Graphite Fiber CN 90 with an identical test setup and observe a nonlinear

behavior. Furthermore, due to the low breakage strain of the high modulus pitch fiber, fiber breakage occurs at lower strain levels, thereby increasing the resistance successively with every load cycle. In subsequent experiments, the authors research group includes the strain sensing fibers into a variety of applications [54]. For example, the authors show that it is possible to measure the deflection of the table of an X-Ray machine by including carbon fiber sensors (Figure 11 (bottom)). In contrast to metallic strain gauges, carbon fiber sensors offer X-ray transparency in this application. The authors furthermore combine multiple carbon fiber sensors to a grid-like structure to measure the strain field of a plate-like structure. From the same research group, Christner et al. [23] analyze the transverse strain sensitivity of the sensor. The authors adhesively bond carbon fiber sensors embedded into a single GFRP ply onto both aluminum and CFRP specimen. They change the effective Poisson ratio of the specimen by varying the composite layups and calculate the longitudinal and transverse strain sensitivity from a linear regression with these different specimen. The resulting longitudinal gauge factor is approximately 1.7 and the transverse strain sensitivity is calculated to be approximately 0.4. The transverse strain sensitivity is furthermore confirmed with the standard procedure described in ASTM E251.

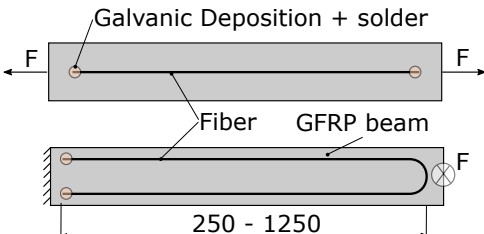

**Figure 11.** Experiments from [23,45].

Kunadt et al. [46,55] similarly analyze carbon fiber sensors for the monitoring of glass fiber reinforced thermoplastic structures. They integrate a HTA40 3 k carbon fiber roving in a meandering manner into the GFRP (Figure 12). They specifically analyze typical measurement errors encountered in practical measurements: errors of linearity, hysteresis, creeping as well as temperature coefficient and repeatability of measurements. For this purpose, the authors use specially designed loading scenarios with increasing and decreasing load of different amplitude to measure the different possible error. The authors contact the carbon fiber by winding a metal wire around the fibers and report a small contact resistance of 20 mΩ. The authors measure a gauge factor of approximately 1.3 with linearity error smaller than 8%, a hysteresis error of approximately 2%, and a creep error smaller than 4%.

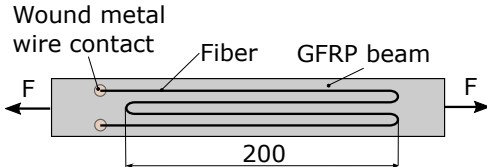

**Figure 12.** Experiments from [46].

Huang et al. [47,56] analyze the electromechanical coupling of the T700SC 12k roving using silver paste as electrodes and a wheatstone bridge for measuring the resistance change (Figure 13). They specifically analyze the influence of different length to width ratios of the CFRP material. The authors argue that, under a tension force, the fibers are more concentrated in the transverse direction then when free from stress, thus leading to a more uniform transverse electrical conduction. The authors argue that this leads to a two stage process: For low strain levels, the random interconnections of the fibers in transverse direction becomes more and more homogeneous, thus reducing the transverse resistance. At a certain threshold, the interconnections reach a stable state, thereby leading

to a nonlinear behavior at low strains and a linear behavior at high strains. In order to prove their hypothesis, the authors analyze specimens with different width to length ratios. They observe, that the linearity of the sensor signal increases with decreasing w/l ratios. For w/l ratios smaller than approx. 0.012, the authors find a linear sensor behavior with a gauge factor of 2.

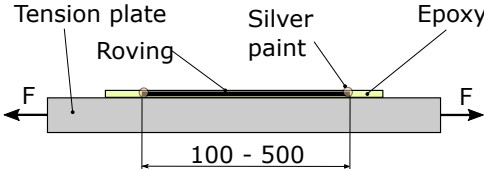

**Figure 13.** Experiment from [47].

In further articles, researchers attempt to influence the sensor behavior of a carbon fiber roving by changing the manufacturing conditions of the sensor. Specifically, some authors analyze the influence of applying different tensile loads to the fiber before and during manufacturing. The idea to use these prestresses during manufacturing is also applied in other research areas to improve mechanical properties of the composite. Mostafa et al. [57] wrote a thorough research article that explains a lot of approaches attempted in this field. The authors report on a number of articles that proved the possibility to increase mechanical properties of fiber reinforced plastics through prestresses. For example, the authors report that fiber waviness can be effectively reduced and that a compressive residual strain can be induced to the matrix that reduces crack propagation in the matrix. It is therefore reasonable to expect that the functionality of the material as a sensor can also be altered by this method.

Yang et al. [22] analyze specifically the influence of pre-tensioning during manufacture of carbon fiber based strain sensors. They analyze a PAN-based carbon fiber with a wheatstone bridge circuit (Figure 14). They apply strains ranging from 0 $\mu\epsilon$ to 200 $\mu\epsilon$ to the roving for at least one day before being coated with resin. The authors show, that a pretensioning of more than 200 $\mu\epsilon$ significantly improves the linearity of the sensor, achieving a correlation coefficient of 0.997 and linear gauge factors of approximately 3. Specimen with less than 100 $\mu\epsilon$ prestrain show an initial decrease of resistance for strains below 0.1% with a subsequent increase with approximately the same slope as higher pre-strained specimen. The authors attribute this phenomenon to an increase in fiber alignment during manufacturing as well as to a reduction of fiber residual compressive stress. When loaded and unloaded multiple times after one another, the authors observe a residual increase of resistance of approximately 0.04% after unloading. They attribute this to initial fiber fractures. In later experiments, the authors further observe a slow decrease in electrical resistance after unloading that lasts for several hours.

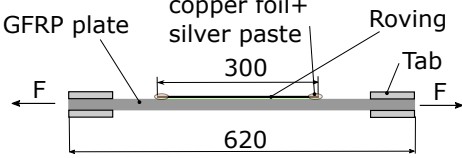

**Figure 14.** Experiment from [22].

Saifelden et al. [49,50] perform experiments on the carbon fiber T700SC with a wheatstone bridge (Figure 15). In [49], the authors use a pretension of 500 $\mu\epsilon$ before resin impregnation for 24 h. They observe a progressive sensor behavior for strain levels of up to 4000 $\mu\epsilon$. The authors argue, that this is due to the still existing initial waviness of the fibers. While they applied a pretension of 500 $\mu\epsilon$ during fabrication of the sensor, the authors argue that still not all fibers can be arranged in a flawless straightness. Thus, for small strain levels, the tension force is not evenly distributed among all fibers and not

all fibers are strained. At higher strain levels, more and more fibers become load carrying, thereby acting as part of the sensor-network. The authors then analyzed a post tensioning process, where the sensors are being loaded under 60% of their failure-load for three hours. The authors argue, that the fibers will thereby straighten due to relaxation of the hardened resin. The authors observe a successive increase in resistance of about 1% until two hours after the load is applied. As a result, the sensors show a linear behavior up to a strain of 6000 $\mu\varepsilon$ when treated with the post-tension procedure. The gauge factor is measured to be approximately 5. In [50] the authors analyze the influence of different sensor lenths between 50 mm and 500 mm and different pre- and post-tensioning levels. As a conclusion, the authors recommend to apply 60% of the ultimate load both before resin impregnation and after hardening.

Höhne et al. [58] analyze a different type of carbon fiber sensor. They use a pitch based high modulus fiber and embed a single roving into an epoxy matrix. The authors apply a so called functionalisation cycle, where the sensor is loaded above its maximum strain to result in broken fibers. When the roving is afterwards loaded, the authors find a very large and reversible resistance increase with slight hysteresis. The authors find, that the fibers break in a zig-zag pattern. When the sensor is loaded, the broken fibers seperate and the current has to be transmitted through different conduction paths. This mechanism results in a nonlinear resistance change over strain with very large gauge factors of more than 2000. While this is an interesting approach by itself, it somehwat contradicts the Self-Sensing philosophy of a structure that is used for both load carrying and strain measurement.

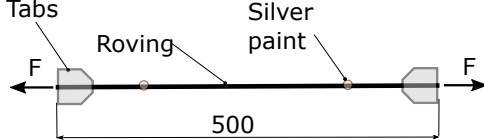

**Figure 15.** Experiment from [50].

**Discussion of roving experiments**

Overall many authors report on gauge factors that are reasonably close to the values reported in single fiber experiments. This supports the interpretation, that the resistance change of rovings embedded in a polymer matrix during longitudinal straining is predominantly effected by the load carrying fibers used in the study. Studies using carbon fiber rovings reported to this point do not analyze ultra high modulus fibers that have shown large negative piezoresistivity in single fiber experiments. We believe an experimental study using ultra high modulus rovings would be very useful to further clarify the piezoresistive response of embedded carbon fiber rovings.

In comparison to many single fiber experiments, nonlinear resistance changes have been more frequently demonstrated. The linearity of sensors has been shown to improve when the sensors are rather long for a given number of fibers [47], and when the fibers are aligned in a very straight manner by stretching them during manufacturing [22,45,50]. Both relatively short specimen and specimen that do not have well-aligned fibers in some cases showed a decrease of resistance for small strains [22,47,50].

Pre-strained carbon fiber sensors in some cases show gauge factors significantly larger than 2 [22,50]. Another study shows that fiber breakages caused by very large stresses to the sensor drastically increase the observed gauge factor [58]. We could hypothesize that the pre- and post-straining cycles—even with forces below the ultimate failure load—also break fibers, thus resulting in these somewhat larger gauge factors compared to other roving experiments.

It has to be noted, that all articles reviewed in this section use 2-wire measurements or wheatstone bridge measurement, that could be unreliable due to the inclusion of contact resistance. However, very similar results are reported even though some experiments use electrical contacts that are mechanically loaded [45] and others position contacts outside

the loaded zone [51]. Furthermore, sensors of different lengths have been studied, many of which showing comparable results. Finally, vastly different contacting methods such as silver paint and galvanic deposition followed by soldering, were used, many of which again giving comparable results. This comparison indicates that 2-wire measurements can be reliable in these cases. This hypothesis however has to be experimentally validated by simultaneous 4-wire and 2-wire experiments on carbon fiber rovings.

## 6. Using large Carbon Fiber Reinforced Plastic Structures as Strain Sensors

### 6.1. Longitudinal Resistance Change with Longitudinal Strain

Instead of using discrete carbon fiber rovings as sensors, a number of authors analyze the suitability of measuring the resistance change of a larger laminate for strain sensing. The electrical conduction processes of these specimen are more complex due to the even larger number of conduction paths within the material. In comparison with experiments on single rovings, current inhomogeneties are likely to influence the measured signal more significantly. It has to be carefully analyzed what role this electrical conduction plays in the experimental results. Furthermore, due to the overall smaller resistance of large specimen, cable and contact resistances will play a more significant role in the experiments if not compensated since they make up a more significant fraction of the overall resistance. Table 4 shows the resulting longitudinal gauge factors reported on experiments using carbon fiber laminates. All of the experimental approaches used in these studies are aimed to measure the same physical property. However, as Table 4 shows, different experimental setups can lead to vastly different observations, even if identical carbon fibers are used within the laminate.

Wang et al. [2] study the electromechanical coupling of the carbon fiber *Torayca T-300* embedded in unidirectional manner in an epoxy matrix with lateral surface contacts, silver paste as electrical contacts and the 4 wire measurement technique (Figure 16). They furthermore measure the transverse resistance change by applying a silver paste electrode on the top and the bottom of the specimen. The authors observe, that both the longitudinal and the transverse resistance change in a partly reversible and partly irreversible manner. The resistance changes nonlinearly with strain, with strain sensitivities of about $-36$ in longitudinal direction and $+40$ in transverse direction. The authors argue, that the fiber alignment increases with tension. Adjacent fiber layers thus have a lower chance of fiber contact under strain, thereby increasing the transverse resistance. Furthermore, the authors argue that the increase of fiber alignment causes the longitudinal resistivity to decrease, thus explaining the negative strain sensitivity. The authors later publish another paper [59] further discussing the subject. The objective of this paper is to further clarify the observed phenomena and specifically to discuss the differences in results when using the Type 1 and Type 2 geometries. They therefore measure four different variants, namely (i) a four-wire surface contact version, (ii) a four-wire combined surface and end contact version(Where the current wires were at the end of the part), (iii) a two-wire end contact specimen and (iv) a two wire surface contact specimen. They analyze the T-300 fiber from Torayca embedded in epoxy. Parts are cured using prepreg material and a heated press. Silver paste is applied as electrical contact to the fibers. The main conclusion of the authors is that 2-wire methods measure the contact resistance rather than the true resistance of the carbon fiber part. The authors further argue, that positive gauge factors measured this way do not reflect the true piezoresistive nature of the material, but rather the degradation of electrical contacts. The 4-wire methods display the true negative piezoresistance of unidirectional CFRP, which is a result of increased fiber alignment. In their most recent paper on this topic Wang et al. [60] argue, that this type of negative piezoresistivity only occurs for specimen that show a large initial fiber waviness. In this article, the authors analyze a quasi-isotropic lamina with 24 plies $[0/45/90/-45]_{3s}$. They conclude, that longitudinal resistance measurements are not suitable for strain monitoring due to the small resistance change and recommend to measure the through-thickness resistance change instead.

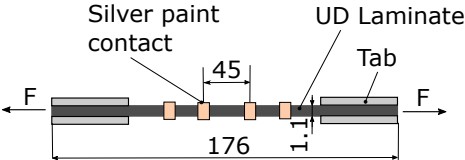

**Figure 16.** Experiment from [59].

Gordon et al. [61] study the difference of experiments performed on single and double ply specimens. They use the PAN based Fortafil 555 carbon fiber for their experiments. Silver paint is used as electrical connection on the lateral surface of the specimen. For current introduction, the matrix is burnt away at the specimen ends and the fibers are coated with the silver paint (Figure 17). Afterwards they are attached to a copper strip through the silver paint and a wire is soldered to the copper strip. The authors observe positive gauge factors of 1.8 to 3.5 for the single ply laminate. For double ply specimen, the authors observe a negative gauge factor of approximately −5.5. The authors thus conclude that the interply region plays an important role to achieve large, negative, gauge factors.

**Table 4.** Literature review of electromechanical coupling of larger carbon fiber parts consisting of multiple rovings.

| Manufacturing | Material | Thickness/mm | Connection | Method | Gauge Factor | Source |
|---|---|---|---|---|---|---|
| prepreg compression moulded | T300/Hy-E1076E | 1.4 | silver paste | 4-wire | −35−−38 | [2] |
| prepreg compression moulded | T300/Hy-E1076E | 1.1 | silver paste | 4-wire | −23 | [59] |
| prepreg compression moulded | T300/Hy-E1076E | 1.1 | silver paste | 2-wire | 3 | [59] |
| prepreg compression moulded | T300/Hy-E1076E | 1.1 | silver paste | 4-wire | −4.6 | [59] |
| prepreg compression moulded | T300/Hy-E1076E | 1.1 | silver paste | 2-wire | 8.6 | [59] |
| prepreg hotpress | Q111/2500 | 0.25 | silver paste | 4-wire | 2 | [62] |
| prepreg hotpress | Q111/2500 | 1.5 | silver paste polished | 4-wire | 2.6 | [62] |
| prepreg hotpress | Q111/2500 | 1.5 | silver paste unpolished | 4-wire | −20 | [62] |
| prepreg autoclave | T300/914 | 2 | silver epoxy | 4-wire | 3.6 | [63] |
| prepreg autoclave | T300/914 | 2 | silver paint | 4-wire | 20.6 | [63] |
| prepreg autoclave | T300/914 | 2 | carbon cement | 4-wire | −89 | [63] |
| prepreg autoclave | T300/914 | 2 | sputtered chromium | 4-wire | 1.75 | [21] |
| prepreg | Fortafil 555/C2002 | 0.15 | silver paint | 4-wire | 1.8–3.5 | [61] |
| prepreg | Fortafil 555/C2002 | 0.31 | silver paint | 4-wire | −6 | [61] |
| prepreg vacuum pressure | TR30S/ Pyrofil380 | 0.23 | copper plating | 4-wire | 2.35 | [64] |
| vacuum press method | T300/HZ9901 | 7 | silver paint | 2-wire | 2.7 to 3.5 | [65] |
| brush application | T300/HZ9901 | 7 | silver paint | 2-wire | 22 to 28 | [65] |
| prepreg autoclave | PYLOFIL 380 | 0.22 | copper electrodeposition | 4-wire | 1.5–2.2 | [66] |

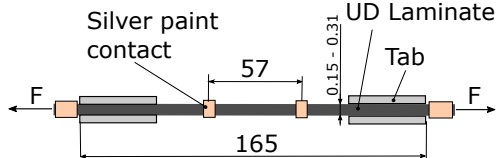

**Figure 17.** Experiment from [61].

Angelidis et al. [21,63] publish the results of experiments using the 4-wire configuration and electrodes made with different materials (Figure 18). They observe a positive gauge factor of 1.7 for the configuration using only surface contacts. They furthermore thoroughly discuss the applicability of different electrode materials. They point out, that carbon cement specifically is not a suitable material due to insufficient contact to the individual fiber. Microscopic images of their specimen showed a degradation of the electrical connection of carbon cement to the part. This resulted in the measurement of a large

negative piezoresistance. The silver paint electrodag and silver filled epoxy on the other hand prove to be a suitable candidate for a stable connection. They also analyze an experiment with surface electrodes and current introduction at the fiber ends and found similar positive piezoresistances of 1.7 only when the current electrodes at the end of the part were glued with a silver epoxy instead of silver paint or carbon cement. In an effort to explain some of the experimental results, the authors discuss the homogeneity of longitudinal current over the cross section. They reason, that, due to the large anisotropy of the material, the current introduction at fiber ends 280 mm apart has to be uniform on a scale significantly less than 0.1 mm. Larger differences in current introduction would directly lead to inhomogeneous current distribution in the cross section, thus strongly affecting the measurement. The findings are further explained in two discussion papers on the topic [67,68].

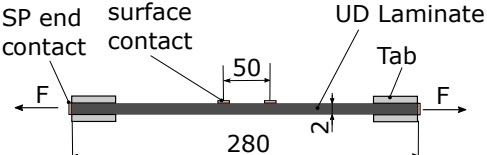

**Figure 18.** Experiment from [63].

Todoroki et al. [15,62,69] analyze surface contact experiments of single ply specimen with a 4-wire measurement and silver paint (Figure 19). Similar to the results of Angelidis et al. [63], the measurements show a positive gauge factor of approx. 2. The authors furthermore perform two types of experiments to show the influence of surface preparation before contacting. In one experiment, the authors thoroughly polish and clean the surface of the specimen. In the second version, the surfaces were not polished. Specimen contacts manufactured without polishing the surface show negative piezoresistivity. The authors explain that this is caused by unreliable electrical contacts.

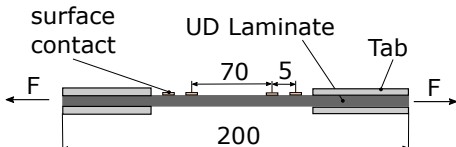

**Figure 19.** Experiment from [69].

Ueda et al. [66] analyze the impact of the inhomogeneous current flow within unidirectional laminates on the measurement of gauge factors via 4-wire tests. The authors analyze a unidirectional specimen with a thickness of 0.22 mm and an anisotropy ratio of $\frac{\rho_T}{\rho_L} = 10e4$ both numerically and experimentally. The specimen has two electrodes mounted 100 mm apart on the top surface (Figure 20). The authors show that the measured gauge factor is dependent on the distance between the two voltage electrodes of the 4 wire measurement. The authors explain, that this is due to the change of resistivity in direction transverse to the fibers. Due to this, the potential distribution in the specimen changes. In their Finite Element studies, the authors show that the measured gauge factor is highly dependent on the distance between voltage measurement and current introduction. In further numerical studies on a specimen with a thickness of 2 mm, the authors were able to show that negative gauge factors are measured when the voltage electrodes are close to the current electrodes and the distance between the current electrodes are smaller than 500 mm.

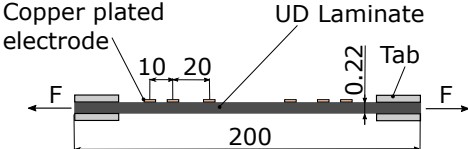

**Figure 20.** Experiment from [66].

**Discussion on laminate experiments**

In summary, there still appears to be some debate on the longitudinal piezoresistivity of carbon fiber laminates. A large, negative piezoresistivity has been independently measured by different research groups, but different explanations for this were put forward.

- Some researchers hypothesize, that the negative piezoresistivity is caused by an increase in fiber alignment and occurs in laminates with relatively large fiber waviness [60]. This argument is similar to other arguments discussed for carbon fiber rovings with ill-aligned fibers [22,50]
- Other explanations invole the interply region of a multy ply laminate [61].
- Again others hypothesize that unreliable electrical contacts could be a reason for large piezoresistivities [63,69].
- Lastly, it has been hypothesized that the change of both longitudinal and transverse resistivity and its interaction with the potential distribution in the cross section can lead to negative gauge factors [66]. It is important to note that this negative gauge factor is measured even though the longitudinal resistance of the fibers grows during straining. The decrease of resistance is not due to a decrease in longitudinal resistance, but results from a change of potential distribution due to the difference in longitudinal and transverse gauge factor. This explanation could therefor explain both the consistently positive gauge factor of roving experiments as well as deviating results from laminate experiments.

More experiments are necessary to clearly answer which of these hypothesis are correct under what circumstances. Overall, current inhomogeneties have been repeatedly mentioned when experimental results of Self-Sensing carbon fiber laminates are discussed. When current is distributed inhomogeneously in the cross section, both longitudinal and transverse piezoresistivity have to be analyzed because current flow cannot be regarded to be 1-dimensional [66]. Since the change in transverse resistivity is thus relevant to the arguments involving current inhomogeneity, the available literature for this resistivity is reviewed in the next section.

### 6.2. Transverse and through Thickness Resistance Change Due to Longitudinal Strain

Notably, both resistance increase and decrease of different magnitudes have been reported.

Todoroki et al. [64] perform 4 wire measurements to measure the change in transverse resistance due to longitudinal strain. Copper is electrodeposited onto the surface of the part as contacting material (Figure 21). The authors find, that the transverse resistance decreases with applied strain in longitudinal direction with a proportionality constant of approx. −2.39. In their experiments, the authors glue CFRP sensors with different electrode configurations onto a unidiretional CFRP base. The base is then loaded in longitudinal tension. Ogi et al. [70] perform somehwat similar experiments but report an increase of transverse resistance with a proportionality constant of 1.45. Instead of bonding a CFRP sensor onto a base, the authors however directly load a contacted laminate in longitudinal strain (Figure 22).

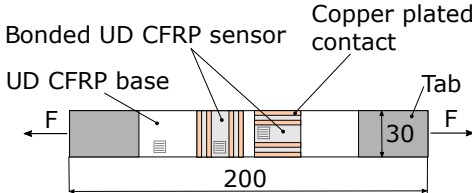

**Figure 21.** Experiment from [64].

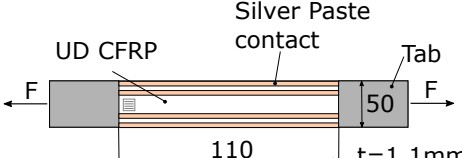

**Figure 22.** Experiment from [70].

Angelidis et al. [21,63] report similar measurements, but use the through thickness direction instead of transverse direction. The authors use sputtered Cr-Au contacts in 2-wire configuration, arguing that a small, reliable contact resistance can be achieved by this means (Figure 23). In their measurements of small longitudinal strains up to 0.3% , the author measure an increase in resistance with a proportionality constant of 2.7. Wang et al. [2] also similarly measure an increase of through thickness resistance upon longitudinal strain (Figure 24) and observe a 2-stage process. The resistance grows rapidly until a longitudinal strain of approximately 0.1% and afterwards grows significantly slower until fracture.

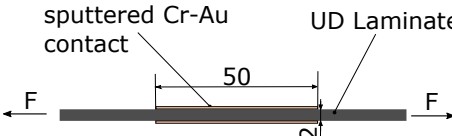

**Figure 23.** Experiment from [63].

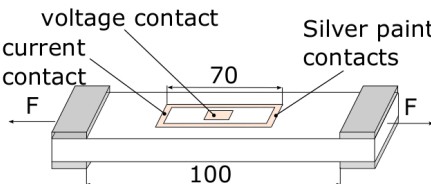

**Figure 24.** Experiment from [2].

Todoroki et al. [64] hypothesize that two main contributing factors change the transverse resistance during longitudinal strain: Increased fiber alignment through longitudinal tension increases the transverse resistance by reducing the number of touching fibers in a given volume. This is often times referred to as the fiber seperation model. Simultaneously, the Poisson effect of the full laminate decreases the overall fiber spacing, and could thereby decrease the transverse resistance. The authors explain, that the magnitude of these mutually competitive effects explain both positive and negative gauge factors in transverse direction.

This is a plausible explanation for the overall results reported on transverse resistance changes. Initial fiber waviness, fiber volume fraction, resin rich interfaces and other factors depending strongly on manufacturing conditions most likely play a significant role in determining how transverse resistance changes due to longitudinal strain. Furthermore

it can be hypothesized that through thickness and transverse resistance change is not the same. Gillet et al. [71] analyze the fiber waviness both in plane and out of plane of an interleaf-toughened CFRP prepreg with microscopy studies. The authors conclude, that the mean in-plane waviness is 3.5 times larger than the mean out-of-plan waviness. It is therefor possible that both transverse directions have different resistivity changes. More experimental and numerical studies that analyze these principles are required to understand the magnitudes of each contributing mechanism and potentially allow to predict which one prevails for a given manufacturing technique.

## 7. Summary, Conclusions and Outlook

The presented literature study shows a large number of research articles working on Self-Strain-Sensing carbon fiber reinforced plastics. Many repeatable experimental results have been reported in the past. However, some conflicting results have also been reported, especially when large and complex specimen are analyzed. Based on the experimental data obtained to this point, we suggest that resistance measurements for Self-Strain-Sensing purposes are most accurate when a homogeneous current flow can be achieved throughout the entire specimen cross section. This can be achieved by using thin and long sensors such as those made from single rovings. Larger specimen built from many rovings are very attractive for Self-Sensing applications, as they can withstand substantial loads. In these cases, a homogeneous current flow can either be guaranteed by reliable contacts at the cut end of the specimen or by using surface contacts with sufficient distance between current and voltage contacts.

In order to increase the comparability of the results between different research groups, it can be derived as a recommendation from the literature analysis to include the following experiments as an accompaniment to future research:

1.  Analyze the electrical and Self Sensing properties of the conductive carbon fiber used in your study with single fiber measurements. Insufficient knowledge of the piezoresistive behavior of the carbon fiber used complicates the interpretation of measurements on more complex structures.
2.  Discuss the current homogeneity of your experimental setup. A good estimation technique is to calculate the rule of mixture resistivity and compare it to the experimentally acquired resistivity. Depending on whether the current density at the points of voltage measurement is larger or smaller than for uniform current conditions, the measured resistivity can also be larger or smaller than the estimated one. If the estimate and the measurement are not reasonably close to one another, a significant current inhomogeneity is present which will have an impact on the experimentally acquired gauge factors.
3.  Report on the surface preparation method used before electrically contacting the carbon fibers. We ascertain that it is necessary to remove any resin rich surface layer to generate a homogeneous current introduction throughout the surface. It has been correctly pointed out in the past that surface polishing damages the fibers [4,60]. However, the removal of a resin rich surface is important for the generation of a uniform current. Other surface preparation methods developed in the field of adhesive technologies such as laser ablation could proove to be usefull for further studies. Chemical etching processes have also been used in the past and may be better suited to remove a resin rich surface without damaging the fibers [66,72]. We believe that the development of repeatable surface preparation technologies in conjunction with reliable electrical contact manufacturing will play a critical role in the further development of Self-Sensing structures.
4.  Report on the resistance change for both the loading and unloading of the sensor to expose potential hysteresis. We suggest loading and unloading the specimen repeatedly multiple times. This allows us to discuss the stability of the sensor during multiple strain cycles. For the analysis of Self-Strain-Sensing properties, the maximum strain should be large enough to obtain results for typical strains occurring in practical



applications, but not so large as to result in irreversible damage. We suggest a maximum strain between 0.3% and 0.5% . Additionally, discuss the linearity of the resistance change with strain. Lastly, report the repeatability of the experiment for multiple specimens fabricated with the same identical manufacturing process.

**Funding:** This research was funded by Deutsche Forschungsgemeinschaft (DFG) Grant No. 447112612.

**Institutional Review Board Statement:** Not applicable.

**Informed Consent Statement:** Not applicable.

**Conflicts of Interest:** The authors declare no conflict of interest.

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
