# Peer review of "A Review on the Usage of Continuous Carbon Fibers for Piezoresistive Self Strain Sensing Fiber Reinforced Plastics"

_jcs, doi:10.3390/jcs5040096_

Round 1
Reviewer 1 Report
This manuscript reviews the carbon fiber-based self-strain polymer sensors. The authors give a systematic investigation on the working principles and characterization of several typical categories of sensors, as well as a thorough discussion about the existing challenges. This work should be of interest to the field. I recommend the publication of this work with some comments listed below.
- This work is focused on carbon fiber based strain sensors. The motivation and unique advantages of using carbon fibers for making strain sensors compared to many other materials should be discussed.
- The application areas of these strain sensors should also be discussed. It would be very nice to have a figure to summarize their applications in a wide range of areas, such as civil engineering, bioengineering, healthcare and smart wearables.
- Recently, multimaterial thermal drawing has evolved as a highly simple and scalable approach for the fabrication of fiber -based strain sensors. This approach exploits the thermal drawing of a macroscopic preform, where functional materials are arranged at a prescribed position, yielding kilometers of fibers with a sophisticated architecture and complex functionalities in a single step. I would suggest including the following publications for a discussion. i) Towards Multimaterial Multifunctional Fibres that See, Hear, Sense and Communicate; ii) Advanced multimaterial electronic and optoelectronic fibers and textiles; iii) Recent Progress and Perspectives of Thermally Drawn Multimaterial Fiber Electronics; iv)Thermally drawn advanced functional fibers: New frontier of flexible electronics.
- The corresponding descriptions of displayed figures are missing in the texts.
Author Response
Dear Sir or Madam,
thank you very much for reviewing our article. Please find below our answers to the points you raised as well as the additional text written in our manuscript.
# Review 1
This manuscript reviews the carbon fiber-based self-strain polymer sensors. The authors give a systematic investigation on the working principles and characterization of several typical categories of sensors, as well as a thorough discussion about the existing challenges. This work should be of interest to the field. I recommend the publication of this work with some comments listed below.
1. This work is focused on carbon fiber based strain sensors. The motivation and unique advantages of using carbon fibers for making strain sensors compared to many other materials should be discussed.
-> We would like to focus the motivation of this research to the idea of Self-Sensing instead of specific material inherent advantages of carbon fibers as a sensor and added a sentence for this in the introduction. We hope that this clarifies the motivation we see behind these ideas
-> "This direct measuring of the load carrying structure and the ideal structural conformity could prove to be advantageous in the future when compared to approaches working with discrete sensors made from a foreign material."
2. The application areas of these strain sensors should also be discussed. It would be very nice to have a figure to summarize their applications in a wide range of areas, such as civil engineering, bioengineering, healthcare and smart wearables.
-> We could not come up with a good way to present a figure that combines all possible applications, but decided to add a few sentences to discuss potential applications. We hope that this nevertheless well sheds light upon possible applications that were proposed in the past.
-> "Possible applications of this idea are manifold and different applications have been proposed in the past. For example, Self-Sensing carbon fibers have been proposed to be used as a health monitoring technology for large bridges\cite{Saifeldeen.2015}, X-Ray transparent deflection monitoring of composite beams\cite{Horoschenkoff.2011} and load and failure detection in composite aircraft wings \cite{Schulte.1989}. Furthermore, resistance measurements have also been proven to be helpful in material testing, where resistance measurements have been used to characterize the failure of single carbon fibers under compression loads\cite{DeTeresa.1991}. "
3. Recently, multimaterial thermal drawing has evolved as a highly simple and scalable approach for the fabrication of fiber -based strain sensors. This approach exploits the thermal drawing of a macroscopic preform, where functional materials are arranged at a prescribed position, yielding kilometers of fibers with a sophisticated architecture and complex functionalities in a single step. I would suggest including the following publications for a discussion. i) Towards Multimaterial Multifunctional Fibres that See, Hear, Sense and Communicate; ii) Advanced multimaterial electronic and optoelectronic fibers and textiles; iii) Recent Progress and Perspectives of Thermally Drawn Multimaterial Fiber Electronics; iv)Thermally drawn advanced functional fibers: New frontier of flexible electronics.
-> Thank you very much for this input. These are indeed very interesting and powerful developments that we were not fully aware of before. We would however like to distinguish it from the general idea of Self-Sensing, since it would involve the integration of a foreign material to a CFRP structure. We decided to add a remark and citation to a very recent article you mentioned and hope you can understand our difficulty to include a large discussion of the topic.
-> "Fiber-based sensors are a popular choice in this field due to their small size and precise measurement. The manufacturing and usage of functional fibers that can fulfill a large number of tasks such as strain sensing, energy storage, energy harvesting and more has been studied extensively and more and more functional principles have been discovered in the past \cite{Yan.2020}."
4. The corresponding descriptions of displayed figures are missing in the texts.
-> We added reference for every picture in the text
Reviewer 2 Report
The paper “A Review on the Usage of Continuous Carbon Fibers for Piezoresistive Self Strain Sensing Fiber Reinforced Plastics” investigated the application of carbon fibers and their reinforced
plastics for Self-Strain-Sensing structures.
The manuscript is interesting; however, to improve the quality, the following recommendations can be incorporated.
* The abstract lacks key findings and contribution of the study to the body of knowledge. Too much background information in the abstract. Put only one or two lines to present the background of the problem and then present the method, findings and contribution of the study.
* In the last paragraph of introduction add final target of this review. The logic of the introduction is not clear enough so that the reviewer is confused to some degree. In order to descript the process and significance of present object clearly, please clarify the technical logic of the present object to improve the introduction.
* The introduction should be rewritten to show the highlights and novelty of the work.
* In the Abstract section, you should describe your work and contribution more clearly than what they were in the background of the research. Currently, it is hard to know main content and results of this article from the Abstract.
* authors should describe about the challenge of this system and add recommendations for the future of it.
* Summary, Conclusion and Outlook part is very long.
* The language of the paper needs to be improved, as such it is really difficult to read...
Author Response
Dear Sir or Madam,
thank you very much for reviewing our article. Please find below our answers to the points you raised as well as the text we added to our manuscript.
The paper “A Review on the Usage of Continuous Carbon Fibers for Piezoresistive Self Strain Sensing Fiber Reinforced Plastics” investigated the application of carbon fibers and their reinforced plastics for Self-Strain-Sensing structures. The manuscript is interesting; however, to improve the quality, the following recommendations can be incorporated.
1. The abstract lacks key findings and contribution of the study to the body of knowledge. Too much background information in the abstract. Put only one or two lines to present the background of the problem and then present the method, findings and contribution of the study.
-> We restructured the abstract and tried to incorporate your ideas
-> Method: "Next, we propose to cluster the available articles into 5 categories based on specimen size and ranging from experiments on bare carbon fiber via impregnated fiber rovings to carbon fiber laminates. Each category is analyzed individually and the potential differences between them are discussed based on experimental evidence found in the past."
-> Findings: "The overview shows, that the choice of carbon fiber and the specific experimental setup both significantly influence the piezoresistive properties measured in Self-Strain-Sensing carbon fiber reinforced plastics."
-> Contribution: We believe our contribution is the systematic study of the different complexity groups and our conclusions and recommendations drawn from this.
2. In the last paragraph of introduction add final target of this review. The logic of the introduction is not clear enough so that the reviewer is confused to some degree. In order to descript the process and significance of present object clearly, please clarify the technical logic of the present object to improve the introduction.
-> We decided to compare our review article to other reviews out there to point out more clearly that this review is aimed to discuss specifically the subject of strain sensing and goes beyond existing work by giving an up-to-date overview as well as a thorough comparison between different experimental setups and specimen sizes. We added this to the end of the introduction.
-> "Other great review articles have been written in the past that discuss a subset of articles analyzed in this work. However, these review articles often work on a broader scope, e.g. reviewing multifunctional polymer-matrix composites in general\cite{Chung.2019,Chung.2012}, or both strain and damage monitoring of CFRP laminates\cite{Todoroki.2007}. This article is dedicated to the specific field of strain sensing in order to more thoroughly discuss research for this application and is aimed to give an up-to-date overview of this field. There is a large number of articles working in this field that have not been communally compared yet, possibly because they work on different types of specimen. For example, some research articles study the piezoresistive properties of single, bare, carbon fibers, while other articles discuss piezoresistive properties of large carbon fiber laminates. We believe that we can learn a lot from analyzing and comparing results between all of these different types of experiments, finding general principles that occur everywhere and trying to explain the reasons for differences in results. To achieve this task, we novelly propose to cluster the existing research into 5 categories of growing part complexity which we believe to be helpful in understanding the functioning principle of a carbon fiber based sensor:"
3. The introduction should be rewritten to show the highlights and novelty of the work.
-> See 2. We hope that the novelty of the work is more clear through the comparison with other reviews we are aware of
4. In the Abstract section, you should describe your work and contribution more clearly than what they were in the background of the research. Currently, it is hard to know main content and results of this article from the Abstract.
-> See 1. We hope we were able to shift the attention a bit more towards our contribution to the research
5. authors should describe about the challenge of this system and add recommendations for the future of it.
6. Summary, Conclusion and Outlook part is very long.
-> We try to discuss experimental challanges thorughout the article, especially when analyzing different experiments such as 2-wire and 4-wire
-> We discuss our proposed path forward through 4 bulletpoints in "7. Summary, Conclusion and Outlook". Since it fits well to the "Outlook" part of our contribution, we hope that you can agree with us that this is a good reason to have the conclusion slightly longer than average.
7. The language of the paper needs to be improved, as such it is really difficult to read...
-> We are sorry to hear you had difficulties to read the article. We re-read the entire article and tried to improve some oddly written parts. We are sure there are more parts that can be improved, but we do not see a way to improve them if they are not pointed out to us specifically
Round 2
Reviewer 2 Report
The article can be accepted for publication.
This manuscript is a resubmission of an earlier submission. The following is a list of the peer review reports and author responses from that submission.
Round 1
Reviewer 1 Report
The authors show review of deals with the application of carbon fibers and their reinforced plastics for self strain sensing structures.
The manuscript presents very basic teóricos approaches, the way the article is written is very tedious due to the large amount of text and does not present many graphs or experimental arrangements illustrating the development of carbon fibers. I consider that the article could be improved by including more graphics.
Reviewer 2 Report
The authors covered an important and up-to-date topic: how the reinforcing carbon fiber can be used for strain sensing function. The increasing number of articles in this field (see: Figure 1.) shows that many researchers have been carried out in recent years and also show the importance of the subject. The cited 76 articles show that they did thorough research in the field.
The novel, stepwise approach of the reviewed articles and organizing those by the “growing part complexity” (see: Figure 2) makes this article interesting and helps researchers to orient themselves on the topic.
However, I have some general and some specific remarks on this article:
- The description of the methodology is not very scientific (line 74-82)
- It is hard to tell, where is the line between literature and own work. For example, the figures are made by the authors based on knowledge derived from the database or the reviewed articles. The syntax (passive voice) and text editing (lack of line breaks) also don’t help to distinguish between the conclusion of the cited authors and the conclusion reached by the authors of this article (e.g. from line 220 or from line 371). It makes more difficult to separate own and cited work that the author’s deductions and experiments are interpolated into the review.
- In some cases, authors are cited, but without citation number (e.g. line 299, line 323-328).
- The footnotes make it difficult to follow the structure, almost in all cases, the information given in the footnote could be written inline in the text with parenthesis (e.g. number 2-4).
- There are too few explanatory figures, especially in chapter 7. More figures of experimental arrangements from the cited articles could help in understanding the topic.
- The authors sometimes repeat themselves (e.g. line 182-184 vs. 187-188).
- The article presented between lines 427 and 435 beside it is interesting, it is not in the scope of this article.
- The SI unit for resistivity is Ωm, in fiber catalogs they use μΩm or Ωmm. To help the comparison of different values available in this article and elsewhere, a more widely used unit should be applied.
- The tables are necessary, but not well structured. The materials are not well defined, as the manufacturer and type should be noted when available. In the first column of Table 3 and Table 4 fiber type, precursor type, filament number, and ply number are mixed and should be written separated.
- Figure 6 and Table 1 show basically the same information from the same data set: how Gauge Factor varies with different Young’s modulus values. The table view gives more understanding of the specific values and the cited articles, thus it should be included in this article after the aforementioned correction.
Reviewer 3 Report
- The subject is suitable for the journal and is in line with current interests.
- This is supposed to be a review paper. However, the authors present their new unpublished results without adequate description of the experimental methods used. For example, the distance between the electrodes and the dimensions of each electrode are unclear in the new results in Sec. 3.3. The company “dpp pultrusion” that made the composite tested is inadequately described. A review paper should not involve a presentation of new results,
- The new results presented involve the two-probe method, even though the Keithley 4-wire in ports are used. The resistance measurement is not reliable using the two-probe method. Thus, the new results presented are not reliable (e.g., Fig. 5-7).
- The literature review is not adequate. The published results are not adequately covered. The prior work obtained under flexure should be included in the review.
- The prior work obtained using the four-probe method should be distinguished from that obtained using the two-probe method. Table 1 lumps both together without distinction. Only the prior work that uses the four-probe method is reliable.
- Please define FVF. Also define “s” in Fig. 5.
- The measured resistivity should not be called the calculated resistivity in the caption of Fig. 5.
- The unit of the resistivity in Fig. 5 is too clumpsy.
- The term “Poisson’s” should have upper case P.
- The term “self-sensing” should be hyphenated.
- This paper is weak in both content and presentation.